# Sensitive Biosensor Based on Shape-Controlled ZnO Nanostructures Grown on Flexible Porous Substrate for Pesticide Detection

**DOI:** 10.3390/s22093522

**Published:** 2022-05-05

**Authors:** Ahmad Fallatah, Nicolas Kuperus, Mohammed Almomtan, Sonal Padalkar

**Affiliations:** 1Department of Mechanical Engineering, Iowa State University, Ames, IA 50011, USA; fallatah@iastate.edu (A.F.); almomtan@iastate.edu (M.A.); 2Space and Aeronautics Research Institute, King Abdulaziz City for Science and Technology, Riyadh 11442, Saudi Arabia; 3Department of Engineering, Dort University, Sioux Center, IA 51250, USA; nicolas.kuperus@outlook.com; 4Microelectronics Research Center, Iowa State University, Ames, IA 50011, USA

**Keywords:** AChE, biosensors, zinc oxide, electrodeposition

## Abstract

Developing an inexpensive, sensitive, and point-of-use biosensor for pesticide detection is becoming an important area in sensing. Such sensors can be used in food packaging, agricultural fields, and environmental monitoring of pesticides. The present investigation has developed a zinc oxide (ZnO)-based biosensor on porous, flexible substrates such as carbon paper and carbon cloth to detect organophosphates such as paraoxon (OP). Here, the influence of morphology and underlying substrate on biosensor performance was studied. The biosensors were fabricated by immobilizing the acetylcholinesterase (AChE) enzyme on ZnO, which is directly grown on the flexible substrates. The ZnO biosensors fabricated on the carbon cloth demonstrated good performance with the detection limit of OP in the range of 0.5 nM–5 µM, higher sensitivity, and greater stability.

## 1. Introduction

In order to satisfy the growing food demands, greater yields in agriculture have become imperative. Consequently, the use of pesticides is crucial in agriculture. Pesticides are widely used worldwide to prevent losses and improve profits in agricultural production. This has also led to indiscriminate use and mishandling of pesticides. Due to the large usage of pesticides, the risk of contaminating food, water, and the environment has also increased. Pesticides such as organophosphates are ubiquitous and need to be monitored regularly. An organophosphate pesticide such as paraoxon, which is a very toxic chemical, has the ability to inhibit the enzyme acetylcholinesterase (AChE). This is one of the most important enzymes in the nervous system of vertebrates, pests, and humans. When exposed to such a pesticide in large doses or for a prolonged period of time, severe health problems can arise such as paralysis, heart attack, stroke, cancer, and even death. These health problems are very prevalent in developing countries, where the agricultural workers are less informed of the ill effects of pesticides. Thus, testing and monitoring pesticides in food, water, and their use in the agricultural fields are very important to mitigate the harmful effects of pesticides [1,2,3,4,5,6,7].

Presently, the standard techniques used for the determination of pesticides are chromatographic methods. These techniques are expensive and require trained personnel to operate and analyze the outcomes. Further, these tools are not portable and require specific operating conditions. Thus, they cannot be utilized as point-of-use devices. The need for a portable sensing device that can be used in food packaging industries, agricultural fields, or at home is imperative [8,9]. To that end, optical fiber sensors and electrochemical sensors are fast becoming appealing tools for sensing and monitoring of such analytes. Many researchers have studied and implanted optical fiber sensors. These sensors have been used in different applications, including the detection of organophosphate compounds, in aquaculture, and per fluorinated compounds [10,11,12,13]. These sensors are small in size, easy to make, and very sensitive. Other advantages of the biosensor are the simplicity, low price, compatibility, and light weight [10,13]. Similarly electrochemical sensors are portable, inexpensive, can be operated by a novice, and provide reliable outcomes [14,15,16,17,18].

Electrochemical sensors have be extensively explored, especially the nanomaterials used in these devices. These nanomaterials range from noble metals to carbon-based materials to metal oxides such as CuO [19], and CeO_2_ [20] and black phosphorus (BP) [17]. There are several metal oxide semiconductors including TiO_2_, ZnO, CuO, SnO_2_, WO_3_, etc., that exhibit promising properties that can be explored for the biosensing applications. In the present investigation, metal oxide nanomaterials, specifically zinc oxide (ZnO), have been used as the biosensing platform. The semiconducting ZnO has been used in applications, namely, optics, electronics, biomedical devices, wound healing, UV filtration, solar cells, varistors, piezoelectric systems, antibacterial agents, photonics, gas sensors, and biosensors [21,22]. It is a wide-band-gap semiconductor, with the conduction band alignment favorable for easy electron transfer. The isoelectric point of ZnO makes it a favorable candidate for biosensing applications. Further, the ease of synthesis of the ZnO nanostructures is very appealing. Moreover, a variety of morphological changes can be created with the help of chemical agents during the synthesis process. The changes in concentration of the chemical agents can further produce morphological and density changes. These advantages over other metal oxides were considered during the process of investigating ZnO as the nanomaterial candidate for biosensing. Furthermore, the wide range of applications demonstrates practicality of ZnO due to its useful material properties [23,24,25,26,27,28,29,30,31,32,33].

In the present investigation, ZnO has been used for the development of a pesticide biosensor that detects trace amounts of paraoxon. Here, the morphology-controlled ZnO nanostructure was directly grown on the underlying substrates via electrodeposition in the presence of morphology-modifying agents. The underlying substrates studied in this work were carbon paper and carbon cloth. The ZnO nanostructures served as a high surface area supporting platform as well as a highly reactive surface that facilitated adsorption, such as AChE enzyme immobilization for pesticide detection. The fabricated ZnO biosensors were extensively assessed for variations in morphology, variability in substrates, catalytic activity, and stability. Our findings indicate that the morphology of the ZnO nanostructures and the underlying substrates significantly influences the biosensor’s inhibition percentage and detection limit.

## 2. Materials and Methods

### 2.1. Chemicals

The chemicals used in the fabrication process include the zinc precursor, zinc nitrate hexahydrate (Zn (NO_3_)_2_·6H_2_O, ≥98%) purchased from Sigma Aldrich (St. Louis, MO, USA). The morphology-modifying agents were ammonium fluoride (NH_4_F, ≥98%), potassium chloride (KCl), and ethylenediamine (EDA, ≥99%) obtained from Sigma Aldrich (St. Louis, MO, USA). The substrates used for electrodeposition were carbon cloth (AvCarb Material Solutions, 1071 HCB) and carbon paper (Toray 060, Wet Proofed), obtained from the Fuel Cell Shop (College Station, TX, USA). The AChE enzyme (200–1000 units/mg), acetylcholine chloride (AChCl, ≥99%), paraoxon-ethyl (C_10_H_14_NO_6_P, ≥90%), and phosphate buffer saline (PBS, pH = 7.4) were used for the development of the biosensor and purchased from Sigma-Aldrich. For cleaning procedures, acetone (100%, 200 proof), hydrochloric acid (HCl, 36.5–38.0%), and nitric acid (HNO_3_, 68.0–70.0%) were used and purchased from Fisher Scientific. Deionized water (DI) was used to prepare all the precursors; however, the analyte and enzyme solutions were prepared using the PBS solution.

### 2.2. Fabrication of ZnO Nanostructures via Electrodeposition

The fabrication process was initiated by cleaning the underlying substrates. The substrates were sheared to obtain 1.25 × 2.5 cm^2^ size pieces. The carbon paper substrate was sonicated in an acetone bath for 10 min, using a Branson 3800 ultrasonic cleaner. It was then cleaned using HCl and HNO_3_ for 1 min each. After each cleaning step, the carbon paper substrate was rinsed with deionized water. The carbon cloth substrate was cleaned in three steps. In the first step, the carbon cloth was immersed in acetone for 1 h. It was then treated to boiling water for 10 min. Finally, it was cleaned in an acetone bath for 10 min using the sonicator. Similar to the carbon paper, the carbon cloth substrate was rinsed with deionized water between each cleaning step [34].

The electrodeposition of ZnO nanostructures was performed using a three-electrode electrochemical cell. Here, the cleaned substrates served as the working electrode. The reference and counter electrodes were Ag/AgCl wire saturated in 1 M KCl and 2 mm diameter platinum wire, respectively. The zinc precursor was the electrolyte with the addition of morphology-modifying agents for obtaining morphology-controlled ZnO nanostructures. The electrodeposition was conducted at −1.0 V applied potential, a temperature of 70 °C, and time of 30 min. The concentration of (Zn (NO_3_)_2_·6H_2_O) aqueous solution was 10 mM. The NH_4_F, KCl, and EDA chemical additives had a concentration of 3 mM, 20 mM, and 2 mM, respectively. After the deposition was complete, the working electrode was rinsed with deionized water, oven dried, and subjected to oxygen plasma treatment (PDC-001, Harrick Plasma, Ithaca, NY, USA). The operating conditions for the plasma treatment were medium radio frequency power, pressure of 500 mTorr, and duration of 1 min. This treatment was performed to improve the wettability of the substrate, which helped in the fabrication of the biosensor. For the present investigation, four samples were prepared. One was a control sample, which was abbreviated as ZnO, and the remaining three samples were prepared in the presence of morphology-modifying agents such as KCl, NH4F, and EDA. These samples were abbreviated as ZnO + KCl, ZnO + NH_4_F, and ZnO + EDA. Henceforth, these samples will be addressed with their abbreviations.

### 2.3. Fabrication of ZnO-Based Biosensors

The biosensor was fabricated on the ZnO nanostructures directly grown on the flexible substrates such as carbon paper and carbon cloth. The ZnO nanostructures with high isoelectric point (IEP 9.5) make it favorable for effective immobilization of biomolecules such as AChE enzyme with a low IEP of 5.35. The immobilization is carried out by dissolving 1 mg of AChE in 1 mL of 10 mM PBS solution (pH = 7.4), followed by immersion of the ZnO nanostructure substrate into the AChE solution for 1 h. The substrate was then rinsed using PBS solution to remove any undetached polymeric chains of the enzyme. The ZnO nanostructure substrate with immobilized AChE enzyme was stored in PBS solution at 4 °C overnight for future use.

### 2.4. Amperometric Measurements on ZnO-Based Biosensors

Upon completion of the biosensor fabrication process, the biosensor was transferred to the electrochemical cell to perform amperometric measurements. For the amperometric measurement, the electrochemical cell consisted of working, counter, and reference electrodes. The electrolyte was the PBS solution at pH = 7.4. The ZnO-based biosensor served as the working electrode. The Pt wire and Ag/AgCl wire saturated in 1 M KCl served as the counter and reference electrodes, respectively. Here, AChCl solution was used as a measure to study the effectiveness of the ZnO-based biosensor. The amperometric measurements were obtained at the oxidation potential of AChCl and were found to be 0.4 V. Upon the addition of AChCl to the electrolyte, the amperometric current was measured. The working electrode was then exposed to OP pesticide by immersing the working electrode in PBS solutions containing different concentrations of OP for 15 min. Following the OP exposure, the working electrode was transferred to the electrochemical cell to obtain further amperometric measurements. The difference in the current response of the working electrode before and after the OP exposure indicated the detrimental effects of OP exposure [35]. The inhibition of the AChE enzyme upon OP exposure was calculated by the following equation.
$$\mathrm{Inhibtion}\;\mathrm{percentage}\;\left(\%\right)=\frac{\left(i_{p,\mathrm{control}}-{\;i}_{p,\exp}\right)}{i_{p,\mathrm{control}}}\times 100$$
where i_p_,_control_ and i_p_,_exp_. are the peak currents obtained from the biosensors before and after OP exposure.

### 2.5. Characterization of ZnO Nanostructures

The crystalline nature and compositional analysis was carried out by performing X-ray diffraction of the prepared ZnO nanostructures. Here, a Siemens D500 instrument was used with a Cu Kα radiation (λ = 1.5406 Å), scan range of 25–65°, and scan step of 0.05° at 45 kV and 30 mA. The absorption properties were studied by using UV–visible (UV–Vis) spectroscopy, using a PerkinElmer Lambda 25 model. The morphologies of the ZnO nanostructure were studied by scanning electron microscopy (SEM) using an FEI Quanta-250 SEM instrument working at accelerating voltage of 10 kV. The electrochemical analyzer (CHI601-CH Instruments) was used to record amperometric responses from the fabricated ZnO-based biosensors.

## 3. Results and Discussion

### 3.1. Crystal Structure, Composition, Absorption, and Morphology Studies of ZnO Nanostructures

The crystal structure and preferred orientation of the fabricated ZnO nanostructures were analyzed via XRD plots. Figure 1a shows the XRD plots of the four samples under investigation. The XRD peaks were indexed and matched well with the wurtzite crystal structure; having JCPDS No. 36-1451 of the powder diffraction pattern [36]. The ZnO sample displayed random orientation with the peak intensities matching with the powder diffraction pattern. However, the remaining three samples prepared with morphology-modifying agents showed preferred orientation along the <0001> direction, with the highest peak intensity for (002) plane. Thus, the XRD plots indicate that in the presence of morphology-modifying agents such as KCl, NH_4_F, and EDA the samples exhibit a preferred orientation, which could be attributed to changes in morphology of the prepared ZnO nanostructures.

XRD data can be used to compute the width of the Bragg peak, which can be used to determine crystallite size. The crystallite size of ZnO nanoparticles was determined using the Debye–Scherrer equation [37].
$$D=\frac{K\lambda }{\beta \mathrm{Cos}\theta }$$
where D is the crystallite size (nm), K is the Scherrer constant, λ is the wavelength of Cu Kα radiation, θ indicates the Bragg peak position, and β is the full width at half maximum intensity of the peak (FWHM). Origin program was used to obtain the diffraction peak broadening (β) from the XRD data. Using the Scherrer method, the average crystallite sizes of ZnO, ZnO + KCl, ZnO + NH_4_F, and ZnO + EDA were determined to be 33.43 nm, 41.58 nm, 41.59 nm, and 55.44 nm, respectively.

Along with XRD analysis, UV–Vis absorption data was collect from the four samples. Figure 1b shows the absorption plots of ZnO, ZnO + KCl, ZnO + NH_4_F, and ZnO + EDA, with absorption peaks at 358, 354, 360, and 360 nm, respectively. The absorption peaks indicate the formation of wurtzite crystal structure of ZnO. The slight variation in the absorption peaks among the samples indicates a change in size and shape of the ZnO nanostructures in the presence of morphology-modifying agents [38,39,40,41].

According to the XRD data, there is a change in the preferred orientation of the ZnO nanostructures in the presence of chemical additives, which could be attributed to morphological changes. Further, the UV–Vis data suggest changes in shape and size based on the changes in the absorption peaks for ZnO samples prepared in the presence of chemical additives. Thus, to confirm these finding of the XRD and UV–Vis data, SEM images were taken for the four samples under investigation. Figure 2 shows SEM images of the ZnO samples. From the images, it is clear that in the presence of chemical additives such as KCl, NH_4_F, and EDA, there is a change in the morphology of the ZnO nanostructures. Figure 2a shows the control ZnO sample, which constitutes hexagonal shape in the nanostructures ~1.2 μm in size. In the presence of KCl, the ZnO nanostructures transform into thin nanosheets or platelets, with a mixture of small and large sizes that range from 3.5 to 7 μm. The thickness of these sheets is from ~85 to ~150 nm. The presence of EDA altered the morphology of the ZnO nanostructures into hexagonal rod-like nanostructures with ~1 μm diameter and a thickness of ~290 nm. The chemical additive NH_4_F also exhibited a change in morphology of ZnO nanostructures into very thin and short nanosheets with an approximate thickness ~50 nm and length ~1 μm. Thus, the changes in the ZnO nanostructure’s shape, size, and density were evident in the presence of morphology-modifying agents such as KCl, NH_4_F, and EDA. The SEM images confirm the findings of XRD and UV–Vis data.

During electrodeposition and in the absence of chemical additives, the zinc ions adsorb on the basal (0001) hexagonal planes. This leads to the formation of the wurtzite crystal structure with no preferred orientation. However, in the presence of chemical additives such as KCl, NH4F, and EDA, the cations in the electrolyte prefer to adsorb on positive polar faces via electrostatic interactions. More specifically, the Cl^−^, F^−^, and NH^2−^ ions adsorb on the positive polar faces and inhibit the absorption of Zn((OH)_4_)^2−^ ions on the polar faces. Due to this preference, an anisotropic growth of ZnO crystals along the <0001> direction occurs. This leads to change in morphology in the presence of chemical additives. The morphology can be further modified with varying the concentration of the chemical additives.

### 3.2. Amperometric Response of ZnO Biosensors to AChCl: Influence of ZnO Morphology

For the amperometric measurements, four samples (ZnO, ZnO + KCl, ZnO + NH_4_F, ZnO + EDA) were investigated and prepared on carbon paper substrates and served as the working electrodes. The amperometric responses for these samples were examined through the successive addition of AChCl. The applied potential for these measurements was 0.4 V. The electrolyte was PBS solution with a concentration of 10 mM and pH of 7.4. Figure 3 shows amperometric measurements obtained from the aforementioned ZnO samples. These amperometric responses were obtained before and after the biosensors were exposed to the pesticide OP.

Before the exposure to OP, the ZnO-based biosensor (Figure 3a) showed a steady increase in current upon the addition, of 50 μL, of AChCl to the PBS electrolyte. A steady-state current was achieved within 2 s of AChCl addition. The observation of a current response was due to the reaction between AChCl and AChE enzyme leading to the formation of thiocholine, which undergoes oxidation to release protons. The chain of reactions is summarized in the following equations.
$$\mathrm{Acetylthiocholine}+H_{2}O\overset{\mathrm{AChE}}{\Rightarrow }\mathrm{thiocholine}+\mathrm{acetic}\;\mathrm{acid}\;\;2\mathrm{thiocholine}-2e^{-}\to \mathrm{dithio}-\mathrm{bis}-\mathrm{choline}+2H^{+}$$

Since these reactions occur on the electrode surface, the properties of the surface play a vital role. Thus, morphology-controlled ZnO nanostructures were used and the influence of morphology change on the biosensor’s performance was studied. The remaining ZnO samples were analyzed for their amperometric response to AChCl. It was observed that the overall current for the remaining samples was greater than the control ZnO sample. This overall increase in current was attributed to the increase in the surface area and an increase in the total number active reaction sites on the surface of the ZnO nanostructures.

Upon exposing the ZnO-based biosensors to the pesticide OP, the current response decreased. This decrease in current was greater with increase in the OP concentration, from 0.001 µM to 5 µM with a detection limit of 0.5 nM. The decrease in the current response was due to the inhibition of AChE enzyme upon exposure to OP. When the ZnO biosensor interacts with OP, the immobilized AChE on the ZnO surface is no longer available to catalyze the hydrolysis reaction of AChCl to thiocholine. This leads to lowering the reaction rate and producing fewer protons upon completion of the reaction. With increase in the OP concentration, the percent inhibition increased, as seen in Figure 8a. However, the percent inhibition values are lower for morphology-controlled ZnO nanostructure samples compared to the control ZnO sample. The ZnO + EDA sample showed the lowest inhibition percentage of 26%. This variation in percent inhibition for morphology-controlled ZnO samples was attributed to the high surface area and greater active reaction sites on the surface. Figure 4 shows the mechanism adopted by the ZnO-based biosensors for the detection of pesticides such as paraoxon.

Along with amperometric responses, electron impedance (EIS) measurements were performed on the ZnO samples. Here, an electrochemical cell with three electrodes was used. The samples served as the working electrodes. The counter and reference electrodes remained unchanged. The reaction area, on the working electrode, was 1 cm × 1 cm. The electrolyte used for these measurements was PBS with a pH of 7.4. Impedance was measured over a frequency range of 0.1 Hz to 100,000 Hz at 0 V with an amplitude of 5 mV.

Figure 5 shows the EIS plots for all four ZnO samples on carbon paper. These plots were obtained at every stage in the detection process. Figure 5a shows the EIS plot for the control ZnO sample. Upon immobilization of AChE enzyme on the ZnO nanostructure surface, the second EIS measurement was obtained. Following immobilization, the ZnO-based biosensor was treated with AChCl, which was added to the PBS electrolyte, and another EIS measurement was obtained. The biosensor was then exposed to the pesticide OP, by immersing the biosensor in an OP solution of desired concentration for 15 min. In the next step, the biosensor was placed back into the electrochemical cell to obtain the last EIS measurement, after the addition of AChCl. From Figure 5a, it was observed that there were two distinct features in the plot. The kinetic controlled region was represented by a semicircle, the diameter of which includes the charge transfer resistance (Rct), while the remaining feature in the plot represented the diffusion-controlled behavior. There was a successive increase in the Rct value after each subsequent step in the detection process. Further, there was greater increase in the Rct value following the exposure to the pesticide OP. This trend was observed in all four ZnO nanostructures.

The value of Rct was attributed to the reaction surface area and the number of active sites on this surface. Thus, the treatment of the biosensor with various substances, namely, AChE, AChCl, and OP, indicates the interaction of these active sites with either the substances or the reaction products of these substances. In the first step, the active sites on the ZnO surface electrostatically interact with the AChE enzyme, which leads to increase in the Rct value. Further, the immobilized AChE catalyzes the hydrolysis reaction of AChCl. This reaction occurs at the AChE site that is electrostatically adsorbed on the active sites of ZnO nanostructures, which leads to further increase in Rct value. Later, when the AChE-immobilized ZnO biosensor is exposed to pesticide OP, the Rct value shows a large increase. This is due to the interaction of OP with AChE, which makes AChE unavailable to catalyze AChCl. Thus, there are fewer active sites on the ZnO surface, with immobilized AChE available to catalyze the hydrolysis reaction of AChCl.

However, when Rct values for morphology-controlled ZnO samples were compared to the control ZnO sample, a decrease in Rct was observed. This decrease in Rct values was attributed to larger surface area and more active sites on the nanostructure’s surface. Table 1 shows the Rct values for the four ZnO samples on carbon paper, at different stages of the detection process.

### 3.3. Amperometric Response of ZnO Biosensors to AChCl: Influence of Underlying Substrate

In this part of the investigation, the influence of the underlying substrate on the biosensor’s performance was studied. Here, carbon cloth was used as a substrate. For this study, the same four ZnO samples, namely, ZnO, ZnO + KCl, ZnO + NH_4_F, and ZnO + EDA, were grown directly on the carbon cloth. Figure 6 shows amperometric measurements obtained from the aforementioned samples before and after the OP exposure. Here, similar trends were observed, as seen in Figure 3. The overall current greatly increased for all samples compared to their carbon paper counterparts. Since carbon cloth constitutes many interwoven carbon fibers forming a three dimensional structure, it greatly increases the reaction surface area and the active sites on the surface. Here, the limit of detection was 0.4 nM and the linear range for detection was found to be 0.001 µM to 5 µM. There was a slight increase in sensitivity for carbon cloth samples. This small increase was attributed to greater surface area and more exposed active sites present on the reaction surface. Among the four ZnO samples, ZnO + EDA samples demonstrated the highest current response and sensitivity.

Interference studies were investigated by another group [42] in the presence of numerous common interfering species to characterize the anti-interference and selectivity of the AChE biosensor. The current signal was measured in the presence of various chemical species and in the presence of harmful chemicals such as methyl parathion (MP), deltamethrin (DM), and Pb^2+^. The biosensors showed no significant change in the current values in the presence of glucose, NO_3_^−^, PO_4_^3−^, SO_4_^2−^, etc. However, in the presence of other pesticides, such as MP and DM, the biosensor showed changes in the current values. Thus, the studied biosensors withstood interference tests effectively and demonstrated good selectivity for the pesticides that were investigated. Our future investigation will perform similar experiments to study interference and selectivity of the prepared biosensors.

Along with amperometric measurements, EIS studies were carried out on ZnO samples grown directly on carbon cloth substrates. Figure 7 shows the Nyquist plots of ZnO control sample, ZnO + KCl, ZnO + NH4F, and ZnO + EDA samples. Similar experimental steps were performed to obtain these EIS plots. From these plots, it is clear that the resistance values increased after the ZnO biosensor was exposed to pesticide OP. Similar trends were observed to those seen in Figure 5. However, a comparison of Rct showed lower values for carbon cloth samples (Table 2). These lower values were attributed to a higher three-dimensional underlying substrate, which increased the reaction surface area. This led to increased number of active reaction cites on the ZnO nanostructures. Further, by virtue of the three-dimensional substrate and the nanomaterial morphology, the path length for the charge carriers was drastically reduced, thus reducing recombination losses and increasing electrical conductivity.

The percent inhibition of the AChE enzyme was plotted as a function of pesticide concentration for ZnO samples fabricated on carbon paper and cloth. Figure 8 show the calibration curves for both carbon paper (Figure 8a) and cloth (Figure 8b) substrates. For carbon paper substrates, the percent inhibition decreased with changes in morphology of ZnO in the following order: ZnO > ZnO + NH4F > ZnO + KCl > ZnO + EDA. The sample ZnO + EDA showed 27% inhibition at a pesticide concentration of 1 µM. Similarly, for carbon cloth substrates, the percent inhibition decreased with changes in morphology. The lowest percent inhibition was obtained for ZnO + EDA sample, which was 26% at 1 µM pesticide concentration of OP. It was observed that 1 µM concentration of OP, all ZnO samples fabricated on carbon cloth exhibited lower percent inhibition compared to their carbon paper counterparts. This decrease in percent inhibition was attributed to increased surface area and increased number of active reaction cites on the biosensor’s surface.

### 3.4. Performance Evaluation of ZnO-Based Biosensors for Detection of OP in Real Samples

For this part of the investigation, ZnO + EDA sample fabricated on carbon cloth was used. Here, the biosensor was evaluated in the presence of real sample, which consists of various complex ions. A fresh biosensor sample of ZnO + EDA was prepared on carbon cloth substrate. Amperometric measurements were recorded for this biosensor before and after pesticide OP (5 µM) exposure. For the first experiment, apple juice was used as the electrolyte and ZnO + EDA sample served as the working electrode. An amperometric response was recorded for successive addition of AChCl in apple juice. In the next step, the apple juice was spiked with OP, and amperometric response of the biosensor was recorded. Figure 9a shows the biosensor performance before and after OP exposure. Here, the current values decreased upon OP exposure. This indicated the presence of OP in the apple juice.

Similar measurements were performed using soil. A measured amount of soil (50 g) was taken and washed with deionized water. This water served as the electrolyte for the amperometric measurement that was carried out on freshly prepared ZnO + EDA sample. In the next step, OP was added to the soil and mixed thoroughly. The soil was then washed with deionized water and used as the electrolyte. The amperometric measurement was recorded after the OP was added to the soil. Figure 9b shows the biosensor performance before and after OP exposure. Upon exposure to OP, the current values of the biosensor reduced greatly, indicating AChE inhibition. From both the amperometric measurements it was clear that the fabricated ZnO-based biosensor was capable of detecting OP in liquid food and in water that passed through OP-contaminated soil. These measurements were promising, and further detailed studies are warranted.

The performance of biosensors in the present work was compared with biosensors studied by other researchers in the field. Table 3 shows important performance parameters of the biosensors studied in the present investigation along with other biosensors from literature. The LOD values of ZnO + EDA samples show promising results. The linear range values were comparable to literature values. This indicates that the fabricated biosensors, especially using carbon cloth substrates, showed promising results and can pave the way for using carbon cloth substrates to improve sensitivity of the biosensor.

## 4. Conclusions

In summary, a comprehensive investigation was carried out to study the influence of ZnO nanostructure morphology and the nature of the underlying substrate to the biosensing performance. This work successfully fabricated morphology-controlled ZnO nanostructures on carbon paper and carbon cloth substrates. The biosensors were prepared by immobilizing AChE on ZnO nanostructures for the detection of pesticides such as OP. The amperometric and EIS measurements demonstrated an overall increase in the current values and decreasing Rct values with increase in the reaction surface area. This indicated a strong influence of nanostructure morphology on the biosensor performance. Further, the carbon-cloth-based biosensors showed better performance, including higher overall current values, higher sensitivity, and lower LOD values, compared to their carbon paper counterparts. These studies showed a greater influence of underlying substrate on the biosensor performance. The biosensors were also capable of detecting OP in complex electrolytes such as apple juice and water passed through OP-contaminated soil.

## Figures and Tables

**Figure 1 sensors-22-03522-f001:**
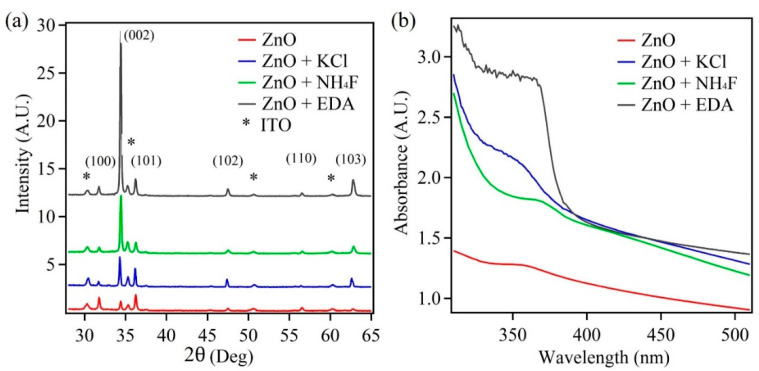
(**a**) XRD plot and (**b**) UV–Vis absorption spectra of the ZnO nanostructures.

**Figure 2 sensors-22-03522-f002:**
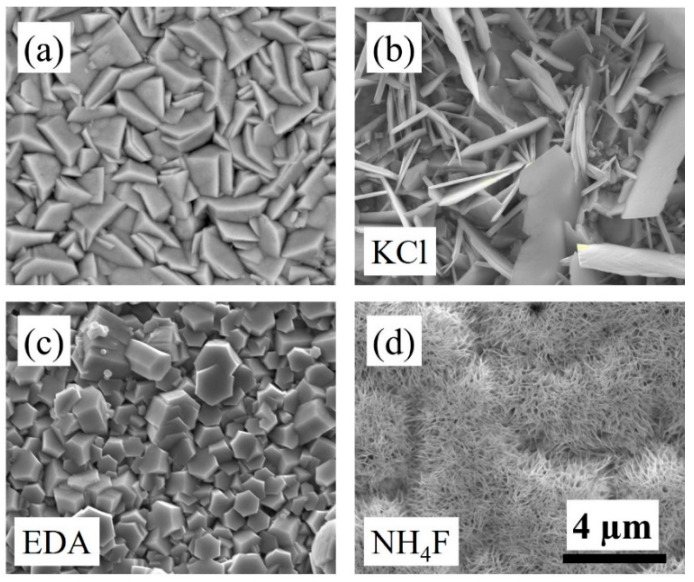
SEM images of electrodeposited (**a**) ZnO, (**b**) ZnO + KCl, (**c**) ZnO + EDA, and (**d**) ZnO + NH_4_F. All images have the same scale bar.

**Figure 3 sensors-22-03522-f003:**
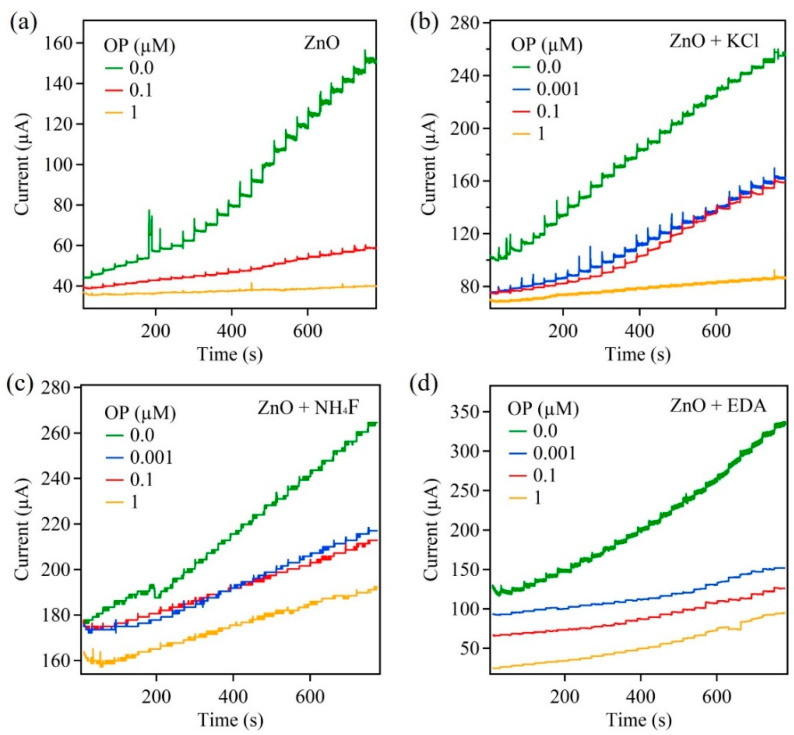
Amperometric response of ZnO samples on carbon paper: (**a**) ZnO, (**b**) ZnO + KCl, (**c**) ZnO + NH_4_F, and (**d**) ZnO + EDA.

**Figure 4 sensors-22-03522-f004:**
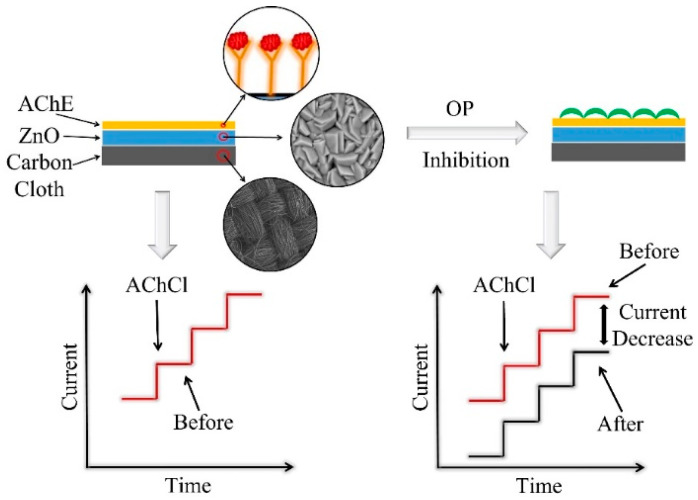
Schematic representing the mechanism adopted by the biosensor for the detection of pesticide.

**Figure 5 sensors-22-03522-f005:**
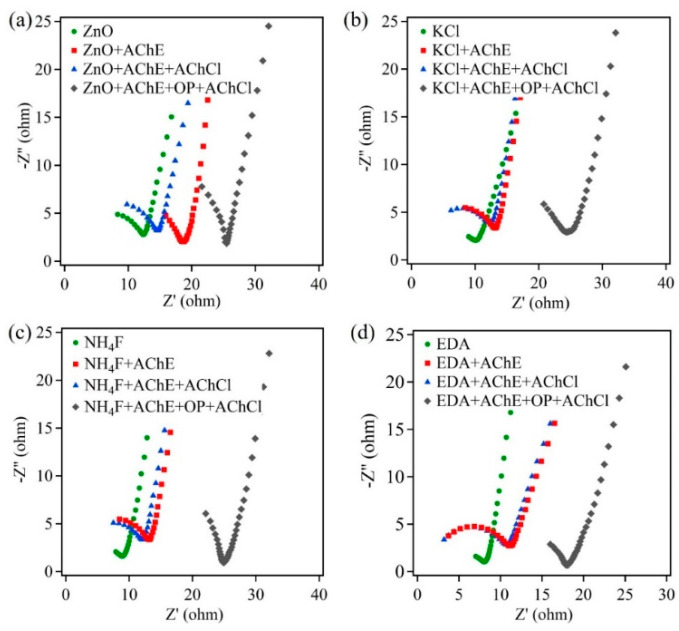
The Nyquist plot of ZnO samples on carbon paper: (**a**) ZnO, (**b**) ZnO + KCl, (**c**) ZnO + NH_4_F, and (**d**) ZnO + EDA.

**Figure 6 sensors-22-03522-f006:**
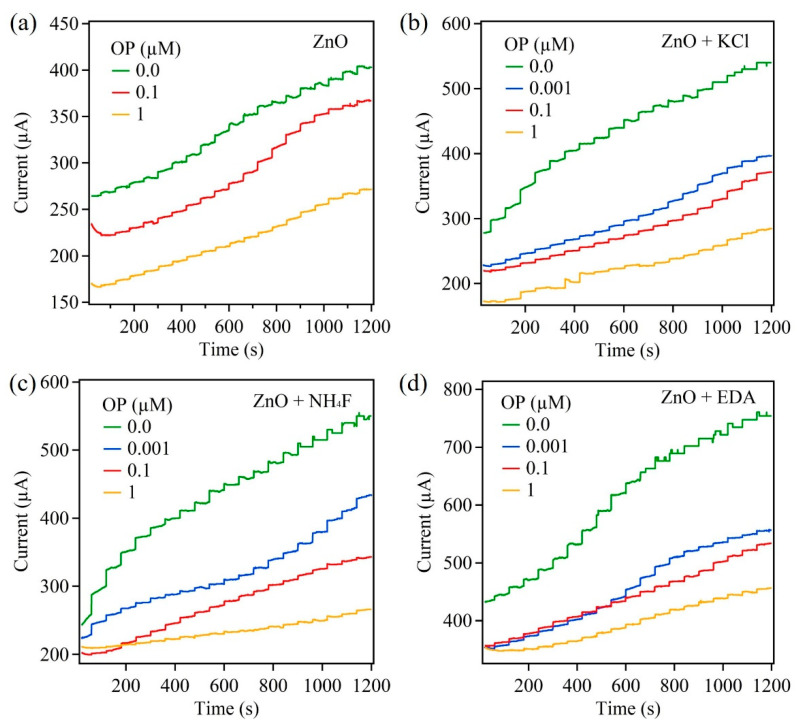
Amperometric response of ZnO samples on carbon cloth: (**a**) ZnO, (**b**) ZnO + KCl, (**c**) ZnO + NH_4_F, and (**d**) ZnO + EDA.

**Figure 7 sensors-22-03522-f007:**
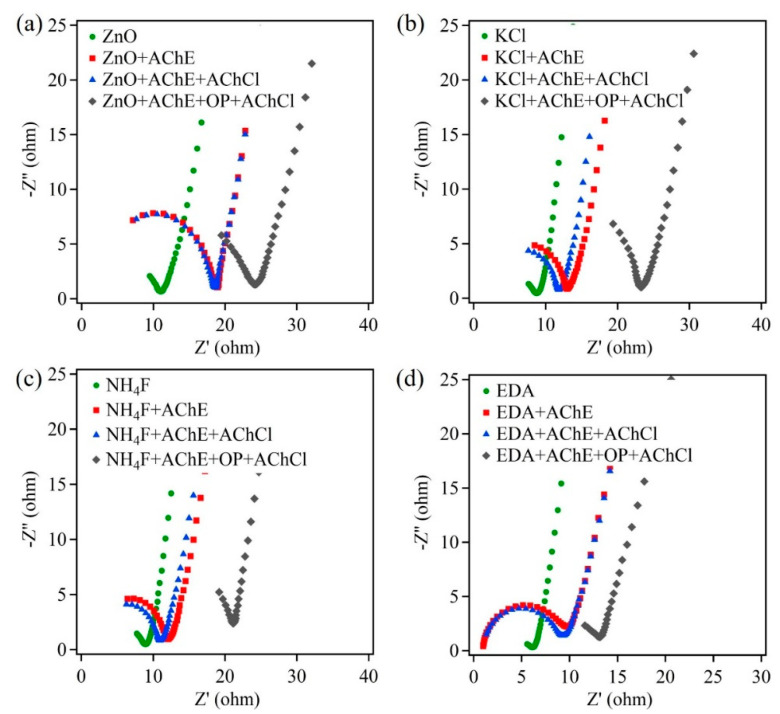
The Nyquist plot of ZnO samples on carbon cloth: (**a**) ZnO, (**b**) ZnO + KCl, (**c**) ZnO + NH_4_F, and (**d**) ZnO + EDA.

**Figure 8 sensors-22-03522-f008:**
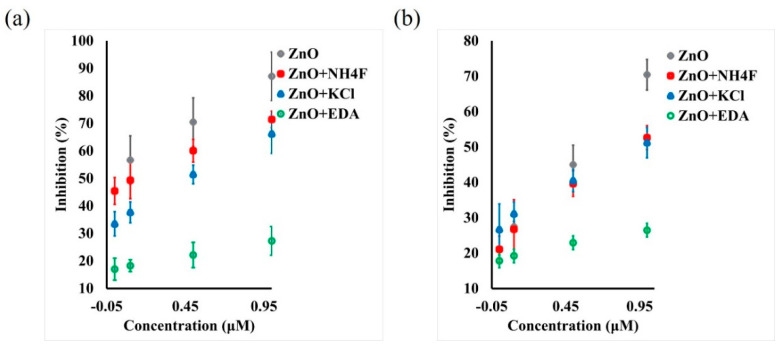
Calibration curve showing paraoxon concentration and percent AChE inhibition on (**a**) carbon paper and (**b**) carbon cloth samples.

**Figure 9 sensors-22-03522-f009:**
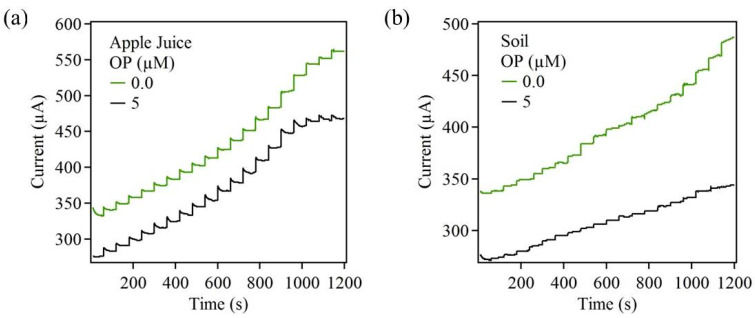
Amperometric response from ZnO + EDA sample on carbon cloth using real samples.

**Table 1 sensors-22-03522-t001:** Rct values obtained from EIS measurements for ZnO samples on carbon paper.

Sample (S)	ZnO (Ω)	ZnO + KCl (Ω)	ZnO + NH_4_F (Ω)	ZnO + EDA (Ω)
S	9.35 ± 1.28	4.27 ± 0.69	3.56 ± 0.66	2.90 ± 0.61
S + AChE	9.56 ± 2.30	10.73 ± 0.81	9.85 ± 1.21	9.42 ± 0.39
S + AChE + ATCl	11.36 ± 1.54	10.55 ± 0.83	10.52 ± 1.37	9.60 ± 0.44
S + AChE + OP + ATCl	18.77 ± 2.47	15.24 ± 2.67	13.88 ± 2.38	11.4 ± 0.69

**Table 2 sensors-22-03522-t002:** Rct values obtained from EIS measurements for ZnO samples on carbon cloth.

Sample (S)	ZnO (Ω)	ZnO + KCl (Ω)	ZnO + NH_4_F (Ω)	ZnO + EDA (Ω)
S	3.67 ± 0.67	2.20 ± 0.37	2.51 ± 0.42	1.10 ± 0.18
S + AChE	10.65 ± 0.29	8.42 ± 0.42	9.09 ± 0.32	8.74 ± 0.18
S + AChE + ATCl	10.38 ± 0.27	8.23 ± 0.64	10.99 ± 0.33	9.90 ± 0.11
S + AChE + OP + ATCl	14.96 ± 1.27	12.3 ± 1.29	12.81 ± 5.69	10.08 ± 0.84

**Table 3 sensors-22-03522-t003:** Performance comparison of present work biosensors with literature values.

Working Electrode	LOD (nM)	Linear Range (µM)	Reference
ZnO + EDA/AChE (Paper)	0.5	0.001–1.0	Present work
ZnO + EDA/AChE (Cloth)	0.4	0.001–1.0	Present work
Pt/ZnO/AChE/Chitosan	0.24	------	[43]
ZnO NPs-CGR-NF/GCE	100	0.2–30.0	[44]
ZnO/CHI/AChE	10	1.75–10.0	[45]
AChE/AuNRs/GCE	0.7	0.001–5.0	[46]
AChE-Carbon Paste/CuE	3.1	0.0–0.12	[47]
Pt/ZnO/AChE	107	0.05–25.0	[48]
Au–PPy–rGO	0.5	0.001–5.0	[49]

## Data Availability

Not applicable.

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
