# Peer review of "Sensitive Biosensor Based on Shape-Controlled ZnO Nanostructures Grown on Flexible Porous Substrate for Pesticide Detection"

_sensors, 2022, doi:10.3390/s22093522_

Round 1

Reviewer 1 Report

The manuscript can be considered for publication in sensors after the following queries are addressed

  1. In the introduction section authors states that "In the present investigation, metal oxide nanomaterials, specifically zinc oxide (ZnO), have been used as the biosensing platform." The reason for choosing  ZnO  over other metal oxides like cuo, W2O3, SnO2 etcg needs  to be elaborated
  2.  in the section 3.1 " the authors have mentioned that "The SEM images confirm the findings of XRD and UV-Vis data" did the authors talk about the particle size comparison in XRD with sem, which parameter from UV vis data is  linked sem and XRD authors need to explain the same and to be included in the revised manuscript.
  3. Did the authors estimate the crystallite size (using Scherer method or any other methods) from the XRD? It would be better if authors calculate and include the same in the manuscript. 
  4. Authors can even explore if possible to include the Raman spectra for these samples so that relation in the crystal structure can be established in comparison with the XRD.
  5. What is the % error in the LOD values ?
  6. The sentence  needs to be corrected "The performance of biosensors in the present work was compared with biosensors studied my other researchers in the field"  as "The performance of biosensors in the present work was compared with biosensors studied by other researchers in the field" 

Reviewer 2 Report

In this study, the authors reported a sensitive biosensor based on shape controlled ZnO nanostructures grown on flexible porous substrate for pesticide detection. The influence of morphology and underlying substrate on biosensor performance was studied. The biosensors were fabricated by immobilizing the acetylcholinesterase (AChE) enzyme on ZnO, which is directly grown on the flexible substrates. The ZnO biosensors fabricated on the carbon cloth demonstrated good performance with the detection limit of OP in the range of 0.5 nM – 5 µM, higher sensitivity, and greater stability. Although being interesting, I find that there are some minor issues with the paper that require addressing prior to this being considered for publication in this journal. I have identified the main points for consideration below:

  1. This manuscript has some spelling typos, style errors and grammatical errors, please correct them.
  2. Some recent reviews relate to the advantages of electrochemical sensors are recommended to be cited, for example Analytica Chimica Acta 1170 (2021) 338480.
  3. A scheme illustration for the sensing mechanism is recommended to be added in the revised manuscript.
  4. The standard plot should be added.
  5. The selectivity of the proposed sensor should be investigated.
  6. The repeatability, reproducibility, and stability of the proposed sensor should be carefully examined.
  7. The caption for Figure 7 is mess.
  8. Error bars should be added in the Figure 7.
  9. Some references related to ZnO-based electrochemical sensors should be cited in the introduction section, such as Surfaces and Interfaces 26 (2021) 101375, Sensors and Actuators B: Chemical, 2021, 346, 130525.

Reviewer 3 Report

This paper reports a potential biosensor based on shape controlled ZnO nanostructures grown on flexible porous substrate for pesticide detection. I have some comments to do before final decision.

  1. Introduction:  "Electrochemical sensors are portable, inexpensive, can be operatedbya46 novice and provide reliable outcomes." -  I miss a critical, short discussion about the optical sensors' advantages compared with electrochemical ones, for such kind of application. Please consider to read and refer the literature that show the advantages of optical fiber sensors for pesticides, agriculture and aquaculture fields: Sensors and Actuators B: Chemical,249, 2017, Pages 523-532; Optics Express 30 (8), 13898-13914, 2022; Talanta 178, 955-961, 2018.
  2. Why this range of measurement?
  3. How about the reproducibility? How many probes were tested? And about reuse? did you test with the same probe, repeat measurements?
  4. How about the selectivity? did you test? Please add more info.
  5. Due to be an electrochemical sensor, how about the robustness for such application to put in the soil or other type of?

Round 2

Reviewer 1 Report

The authors have addressed all the comments. Hence can be accepted for publication.

Reviewer 2 Report

The authors have made substantial changes to the manuscript, improving the resulting presentation measurably. However, I find that there are some minor issues with the paper that require addressing prior to this being considered for publication in this journal.

  1. The selectivity of the prepared biosensor was not investigated in the present work. The authors cited similar work from the literature, and then anticipated similar performance by their biosensors. I don't think that's appropriate. So, I recommend the authors remove this section from the manuscript.
  2. A recent reference (Microchemical Journal 179 (2022) 107515) related to transition metal-based electrochemical sensor is recommended to be cited.

Reviewer 3 Report

Last comment after good revision. When talk bout refs 10 to 13, please add more important literature about Aquaculture parameters like stress hormone, quite important and low cost solution: Biotechnology Reports 29, e00587, 2021.
